

# Investigation into the effects of antioxidant-rich extract of *Tamarindus indica* leaf on antioxidant enzyme activities, oxidative stress and gene expression profiles in HepG2 cells

Nurhanani Razali, Azlina Abdul Aziz, Chor Yin Lim and Sarni Mat Junit

Department of Molecular Medicine, Faculty of Medicine, University of Malaya, Kuala Lumpur, Malaysia

## ABSTRACT

The leaf extract of *Tamarindus indica* L. (*T. indica*) had been reported to possess high phenolic content and showed high antioxidant activities. In this study, the effects of the antioxidant-rich leaf extract of the *T. indica* on lipid peroxidation, antioxidant enzyme activities, $H_2O_2$-induced ROS production and gene expression patterns were investigated in liver HepG2 cells. Lipid peroxidation and ROS production were inhibited and the activity of antioxidant enzymes superoxide dismutase, catalase and glutathione peroxidase was enhanced when the cells were treated with the antioxidant-rich leaf extract. cDNA microarray analysis revealed that 207 genes were significantly regulated by at least 1.5-fold ($p < 0.05$) in cells treated with the antioxidant-rich leaf extract. The expression of *KNG1, SERPINC1, SERPIND1, SERPINE1, FGG, FGA, MVK, DHCR24, CYP24A1, ALDH6A1, EPHX1* and *LEAP2* were amongst the highly regulated. When the significantly regulated genes were analyzed using Ingenuity Pathway Analysis software, "Lipid Metabolism, Small Molecule Biochemistry, Hematological Disease" was the top biological network affected by the leaf extract, with a score of 36. The top predicted canonical pathway affected by the leaf extract was the coagulation system ($P < 2.80 \times 10^{-6}$) followed by the superpathway of cholesterol biosynthesis ($P < 2.17 \times 10^{-4}$), intrinsic prothrombin pathway ($P < 2.92 \times 10^{-4}$), Immune Protection/Antimicrobial Response ($P < 2.28 \times 10^{-3}$) and xenobiotic metabolism signaling ($P < 2.41 \times 10^{-3}$). The antioxidant-rich leaf extract of *T. indica* also altered the expression of proteins that are involved in the Coagulation System and the Intrinsic Prothrombin Activation Pathway (KNG1, SERPINE1, FGG), Superpathway of Cholesterol Biosynthesis (MVK), Immune protection/antimicrobial response (IFNGR1, LEAP2, ANXA3 and MX1) and Xenobiotic Metabolism Signaling (ALDH6A1, ADH6). In conclusion, the antioxidant-rich leaf extract of *T. indica* inhibited lipid peroxidation and ROS production, enhanced antioxidant enzyme activities and significantly regulated the expression of genes and proteins involved with consequential impact on the coagulation system, cholesterol biosynthesis, xenobiotic metabolism signaling and antimicrobial response.

Corresponding author
Sarni Mat Junit, sarni@um.edu.my

## INTRODUCTION

Research on the role of nutrients on general well-being and disease prevention has gained momentum in recent years. Amongst the nutrients studied, natural sources from plants have been gaining notable attention. Plants and herbs have been widely consumed in many populations of the world mainly as food and for various medicinal benefits. Amongst the multitude of bioactive compounds found in plants and herbs, phenolic compounds that are secondary metabolites, have been widely reported as potent antioxidants (*Covas et al., 2006*; *Thangapazham, Passi & Maheshwari, 2007*; *Singh et al., 2008*). Numerous findings support the role of phenolics in the prevention of oxidative damage-related diseases including cancers (*Le Marchand, 2002*), cardiovascular diseases (CVD) (*Reaven et al., 1993*), osteoporosis and neurodegenerative diseases (*Halliwell, 1994*).

*Tamarindus indica* L. (*T. indica*) of which the fruit pulp is widely used as an acidic flavor in cooking, is one of the plants with reported therapeutic properties. *T. indica* belongs to the family of Leguminosae and grows naturally in many tropical and sub-tropical regions. Our group had recently reported that the methanol seed, leaf, leaf veins, fruit pulp and skin extracts of *T. indica* possessed high phenolic content and antioxidant activities (*Razali et al., 2012*). Amongst these extracts, the methanol leaf extract showed the highest antioxidant activities. The leaves are traditionally used to treat various ailments including cough, worm infection, rheumatism, jaundice and ulcer (*Sreelekha et al., 1993*). The methanol leaf extract of *T. indica* has been shown to have antibacterial (*Muthu, Nandakumar & Rao, 2005*; *Meléndez & Capriles, 2006*), antimalarial (*Asase et al., 2005*), antimicrobial (*Escalona-Arranz et al., 2014*), antiviral (*El Siddig, Ebert & Luedders, 1999*), anticancer (*Saleem, 2009*), anti-inflammatory (*Bhadoriya et al., 2011*), hepatoprotective and antioxidant (*Sudhahar et al., 2007*) activities as well as inhibitory effects on tyrosine phosphatase 1B (*Na et al., 2009*). The presence of lupanone and lupeol (*Imam et al., 2007*), catechin, epicatechin, quercetin and isorhamnetin (*Razali et al., 2012*) in the leaf extract could have contributed towards the diverse range of the medicinal activities. A recent study reported that the leaf extract of *T. indica* protected the red blood cells by attenuating $H_2O_2$-induced membrane damage and also inhibiting intracellular ROS production (*Escalona-Arranz et al., 2014*). Molecular evidence to support the beneficial effects of the leaf extract is however, still lacking.

HepG2 cells have long been used as an *in vitro* model to study cytoprotective, genotoxic and antigenotoxic effects of compounds since they retain many of the specialized functions of normal human hepatocytes (*Knasmüller et al., 2004*; *Mersch-Sundermann et al., 2004*). In addition, HepG2 cells have been shown to have the closest similarity in terms of signaling network patterns with those observed in primary hepatocytes (*Saez-Rodriguez et al., 2011*). HepG2 cells have also been used in nutrigenomics analyses including germinated brown rice (*Imam & Ismail, 2013*), *T. indica* fruit pulp (*Razali, Aziz & Junit, 2010*) and *Anacardium occidentale* (*Khaleghi et al., 2011*). Hence, in this study, the effects of the antioxidant-rich methanol leaf extract of *T. indica* on gene expression profile and protein abundance in HepG2 cells, a widely used *in vitro* model for human liver hepatocytes, were investigated.
## MATERIALS AND METHODS

### Chemicals

All reagents used in the experiments were of analytical grade and obtained mostly from Fluka and Sigma. Solvents used for extraction of plant samples were purchased from Fisher Scientific. The phenolic standards were obtained from Sigma. Water used was of Millipore quality. The human hepatoma, HepG2 cell lines were purchased from ATCC, USA. The growth medium, serum and antibiotics for the cell culture experiments were purchased from Flowlab, Australia.

### Preparation of the methanol leaf extract of *T. indica*

The leaves of *T. indica* were collected from Kedah in the northern region of Malaysia and were processed within a day of harvesting. The plant was deposited in the Rimba Ilmu Herbarium, University of Malaya with a voucher specimen of KLU 45976. The methanol leaf extract of *T. indica* was prepared as previously described (*Razali et al., 2012*). Briefly, the dried powdered samples were extracted with methanol at room temperature for 24 h with a mass to volume ratio of 1:20 (g/mL). The extracts were evaporated to dryness on the rotary evaporator at 37 °C and the residues were re-dissolved in 10% DMSO. Extracts were kept at −20 °C until further analyses.

### Cell culture

The human liver cells line, HepG2 (ATCC, Manassas, VA, USA) were grown in Dulbecco's modified Eagle's medium (DMEM) supplemented with 10% foetal bovine serum (Flowlab, Australia), 1% penicillin (Flowlab, Sydney, Australia) and 1% streptomycin (Flowlab, Sydney, Australia). Cells were maintained in humidified air with 5% $CO_2$ at 37 °C.

### Cell viability analysis using the MTT assay

Cell viability assay was carried out using 3-(4,5-dimethylthiazol-2-yl)-2,5-diphenyltetrazolium bromide (MTT) as described by *Mosmann (1983)*, with minor modifications (*Denizot & Lang, 1986*; *Hansen, Nielsen & Berg, 1989*). Briefly, HepG2 cells at a density of 5,000 cells per well were seeded in a 96-well ELISA microplate. The cells were incubated at 37 °C in 5% $CO_2$ for 24 h. After 24 h, the plant extracts, at various concentrations (0–600 μg/ml) were added into the wells and the cells were further incubated for 48 h. After 48 h, MTT reagent (Merck, Kenilworth, New Jersey, USA) was added and the mixture was incubated for 4 h. Then, the mixtures in each well were removed and formazan crystals formed were dissolved in 75% isopropanol. Spectrophotometry measurement of the mixture was performed using a microplate-reader (Bio-Rad, Hercules, California, USA) at wavelengths of 570 and 620 nm. A log plot of cell viability (%) against the concentrations of plant extracts was constructed. From the plot, a near non-toxic concentration ($IC_{20}$) of the extracts was calculated and used for the subsequent analyses on antioxidant activities and gene expression patterns in HepG2 cells treated with the leaf extract. The concentration of the leaf extract that reduced cell viability by 50% ($IC_{50}$) was also calculated from the plot (*Razali, Aziz & Junit, 2010*).

## Evaluation of lipid peroxidation and antioxidant enzyme activities

HepG2 cells were seeded and pre-treated with the $IC_{20}$ concentration of the plant extracts. Following the pre-treatment with the extracts, HepG2 cells were exposed to oxidative stress condition by induction with 1 mM of $H_2O_2$ for 2 h. After 24 h, the cells were subsequently washed with ice-cold PBS and detached using a scraper. Cells were then collected into a microtube and centrifuged for 10 min at 8,000 × g at 4 °C. The supernatant was discarded and the cell pellets were lyzed in Tris–HCl buffer (25 mM, pH 7.4). The lyzed cells were then sonicated for 5 min at 60% amplitude. Protein content of each sample was quantified using Bradford assay, employing bovine serum albumin (BSA) as the standard (*Ausubel et al., 1999*). Hundred μg/ml of protein was used in the lipid peroxidation and antioxidant enzyme assays.

## Inhibition of lipid peroxidation

Levels of lipid peroxidation in the treated cells were determined by measuring the production of malondialdehyde (MDA) in the presence of thiobarbituric acid (TBA) (*Inal, Kanbak & Sunal, 2001*). The TBA reagent used in this assay consisted of a mixture of TBA, trichloroacetic acid (TCA) and 70% perchloric acid ($HClO_4$) in distilled water. Tetraethoxypropane (TEP) in ethanol was used as the standard. Briefly, 0.5 ml of TBA reagent was added to 0.1 ml sample or standard and boiled for 20 min. The mixture was left to cool and was centrifuged for 10 min at 960 × g at 25 °C. The supernatant was pipetted into a 96-well plate and absorbance was measured at 532 nm on a microplate reader. All experiments were done in triplicate. The amounts of MDA in the treated and untreated samples were determined using the equation obtained from the standard curve of TEP. Results were expressed as nmol MDA/mg of protein (*Awah & Verla, 2010*).

## Measurement of HNE-protein adduct

The abundance of 4-hydroxynonenal (4-HNE) protein adducts in both untreated and leaf-treated cells were measured using ELISA kits purchased from Shanghai Qayee, China. Supernatants were collected from the samples and the abundance of protein was detected against 4-HNE (QY-E05206) according to the manufacturer's protocols. Briefly, 10 μl of samples were pipetted into a pre-coated ELISA microplate. The 4-HNE-protein adducts content was probed with an anti HNE-His antibody and was measured by an HRP-conjugated secondary antibody for detection. The plate was washed to remove unbound substances and the end product was measured at 450 nm. A known HNE-BSA standard curve was constructed to determine the concentration of 4-HNE protein adducts present in the samples.

## Antioxidant enzyme activities

Activities of the antioxidant enzymes superoxide dismutase (SOD), catalase (CAT) and glutathione peroxidase (GPx) were measured using Assay Kits purchased from Cayman Chemicals (Ann Arbor, Michigan, USA). The SOD assay utilized tetrazolium salt for detection of superoxide radicals generated by xanthine oxidase and hypoxanthine. SOD

activities were expressed as Unit/ml where one unit is defined as the amount of enzyme needed to exhibit 50% dismutation of the superoxide radicals.

CAT assay measures the reaction of the enzyme in the presence of an optimal concentration of $H_2O_2$. The reaction produced formaldehyde, which was measured colorimetrically with 4-amino-3-hydrazino-5-mercapto-1,2,4-triazole (Purpald) as the chromogen. Purpald specifically forms a bicyclic heterocycle with aldehydes, which upon oxidation changes from colorless to purple. Formaldehyde concentration in the cells was measured using equation obtained from the linear regression of the standard curve. CAT activities in the samples were expressed in nmol $H_2O_2$ oxidized/min/ml of protein.

GPx activity was measured through a coupled reaction with glutathione reductase whereby GPx catalyzes the reduction of hydroperoxides, including $H_2O_2$ by reduced glutathione. GPx activities in the cells were initially calculated by determining the changes in absorbance per minute obtained from the standard curve. One unit of activity is defined as the amount of enzyme that caused the oxidation of 1.0 nmol of NADPH to $NADP^+$ per minute.

## Assay for reactive oxygen species (ROS)

ROS production was assessed using the method of *Wang & Joseph (1999)* with minor modifications. HepG2 cells were plated into black 96-well plates, seeded and treated as previously described. At the end of the treatment, the medium was removed and the cells were washed with PBS and then incubated with 100 mM dichlorofluorescein diacetate (DCF-DA) in PBS for 30 min at 37 °C. The formation of the fluorescent-oxidized derivative of DCF-DA was monitored at emission wavelength of 530 nm and excitation wavelength of 485 nm in a fluorescence multi-detection reader (Synergy HTTM Multi-detection microplate reader; BioTek Instruments Inc., Vermont, USA). ROS production was expressed as unit of fluorescence (UF) produced by DCF-DA/mg of total protein ($UF \times 10^4$/mg of proteins) (*Marabini et al., 2006*).

## Gene expression analyses in HepG2 cells treated with the methanol leaf extract of *T. indica*

HepG2 cells were seeded at a density of $3.0 \times 10^6$ in a 25 cm$^2$ flask. The cells were then treated with an $IC_{20}$ concentration of the methanol leaf extract of *T. indica* for 24 h. Following this, cells were trypsinized and then precipitated by centrifugation at $130 \times g$ for 5 min. Cells were washed with PBS twice before total cellular RNA (tcRNA) was extracted from the cells. tcRNA from both untreated and *T. indica* leaf-treated HepG2 cells was isolated and then purified using RNAEasy kit (Qiagen, Venlo, Netherlands) according to the manufacturer's instructions. The quality and the concentration of the extracted RNA was determined using NanoDrop 8000 spectrophotometer (Thermo Scientific, Newark, Delaware, USA). Only samples with a 260/280 ratio of above 1.8 indicative of good quality, were used for further analyses. The integrity of the tcRNA was assessed using RNA 6000 Nano LabChip kit in an Agilent 2100 Bioanalyzer (Agilent Technologies, Santa Clara, California, USA) as recommended by the manufacturer. Data were archived automatically in the form of electropherogram, gel-like images and tables containing sample concentration and the 28S/18S ratio of the ribosomal subunits. Eukaryotic total

cellular RNA integrity calculated using RIN software algorithm, follows a scale of 1 to 10, with 1 being the most degraded profile and 10 being the most intact (*Schroeder et al., 2006*).

## cDNA microarray analysis

Gene expression analysis was performed on the Affymetrix Human Gene 1.0 S.T (sense target) arrays according to the Affymetrix eukaryotic RNA labelling protocol (Affymetrix, Santa Clara, California, USA). Briefly, the tcRNA (100 ng) extracted from the untreated and the *T. indica* leaf-treated HepG2 cells was first converted to single-stranded sense strand DNA (cDNA) in two cycles using the whole transcript (WT) cDNA synthesis, amplification kit and sample clean-up module. The sense strand DNA was then fragmented, biotinylated and hybridized to the Affymetrix Human Gene 1.0 S.T array at 45 °C for 16 h in hybridization Oven 640. After hybridization, the arrays were stained and then washed in the Affymetrix Fluidics Station 450 under standard conditions. The stained arrays were then scanned at 532 nm using an Affymetrix GeneChip Scanner 3000, and CEL files for each array were generated using the Affymetrix Gene-Chip Operating Software (GCOS).

## Microarray data normalization and analyses

Partek Genomic Suite software was used to convert the CEL files into text files, thus listing all the up-regulated and down-regulated genes in response to the treatment. The gene set annotation was verified using Netaffx Analysis Center software. The gene sets were then subjected to a one-way analysis of variance (ANOVA) in the Partek Genomic software to determine significantly expressed sets of genes, which was set according to $P$ value less than 0.05 ($P < 0.05$). Significantly expressed genes were then re-filtered to include only those with fold change difference of equal to or greater than 1.5. Following the filtration of the genes list, those significant genes were subjected to Gene Ontology (GO) Enrichment tool in the Partek Genomic Suite Software for additional information and further classification on the biological functions of the genes and the genes products. Information on function of genes can be derived from the Gene Ontology database, which provides a structured annotation of genes with respect to molecular function, biological process and cellular component. Further information on GO could be retrieved from http://www.geneontology.org/. All or some selected high significantly expressed genes were then clustered and grouped in hierarchical clustering using Genesis software (*Sturn, Quackenbush & Trajanoski, 2002*).

## Pathway interactions and biological process analyses

Functional analyses to predict networks that are affected by the differentially expressed proteins were carried out using Ingenuity Pathways Analysis (IPA) software (Ingenuity® Systems, http://www.ingenuity.com/). Details of the proteins (encoded by the genes of which the expression is significantly altered), their quantitative expression values (fold change difference of at least 1.5) and $p$ value ($p < 0.05$) were imported into the IPA software to predict biochemical networks that are affected in HepG2 cells in response to the leaf extract of *T. indica* treatment. Each protein identifier was mapped to its corresponding protein object and was overlaid onto a global molecular network developed from information contained in the Ingenuity Knowledge Base. Network predictions based

on the protein input were generated algorithmically by utilizing the information contained in the Ingenuity Knowledge Base. Right-tailed Fischer's exact test was used to calculate a *p* value indicating the probability that each biological function assigned to the network is due to chance alone.

## Validation of the microarray data using real-time RT–PCR (qRT-PCR)

Reverse transcription of tcRNA (1,000 ng) into complementary DNA (cDNA) was performed using a High Capacity RNA-to-cDNA Master Mix (Applied Biosystems, Carlsbad, California, USA) following the manufacturer's instructions. The reaction mixture consisted of 1,000 ng tcRNA and 1 X Master Mix which contained magnesium chloride ($MgCl_2$), deoxyribonucleotide triphosphate (dNTPs), recombinant RNase inhibitor protein, reverse transcriptase, random primers, oligo(dT) primer and stabilizers, was prepared in a final volume of 20 µl. The reaction mixture was placed in a thermal cycler with the following parameters: 25 °C for 5 min, 42 °C for 30 min and 85 °C for 5 min. The cDNA was kept at −80 °C until further analysis.

Primer pairs for selected up-regulated and down-regulated genes as well as a housekeeping gene, *GADPH*, are listed in Table S1. PCR amplification was performed in 0.2 ml MicroAmp® Optical 8-tube strips in a final volume of 20 µl, consisted of 10 µl of 2X Fast SYBR® Green Master Mix (Applied Biosystem, Carlsbad, Calfornia, USA) containing a mixture of SYBR® Green I dye, AmpliTaq® Fast DNA Polymerase, UP (Ultra-Pure), Uracil-DNA Glycosylase (UDG), ROXTM Passive Reference dye, dNTPs and optimized buffer components, reverse and forward primers (200 nm) and cDNA (20 ng). qRT-PCR was performed with a PCR parameters consisted of 20 s of *Taq* DNA Polymerase activation at 95 °C, followed by 40 cycles of denaturation at 95 °C for 3 s, primer annealing at 60 °C for 30 s. The melting curves of the real-time PCR products were analyzed from 65 °C to 95 °C. Each measurement was carried out in triplicate. Differences in gene expression, expressed as fold-change, were calculated using the $2^{-\Delta\Delta Ct}$ method where *GAPDH* was used as the reference gene.

## Detection of altered protein expression in HepG2 cells treated with the methanol leaf extract of *T. indica*

Ten proteins, IFNGR1, LEAP2, SERPINE1, MX1, KNG1, MVK, FGG ANXA3, ALDH6A1 and ADH6 of which the encoding genes were aberrantly expressed in the *T. indica* leaf-treated cells were selected for ELISA analyses. The HepG2 cells ($3 \times 10^6$) were seeded and treated with an $IC_{20}$ concentration of the methanol leaf extract of *T. indica* for 24 h. After 24 h, cell lysates were scraped off from the flasks and re-suspended in ice-cold PBS (pH 7.4) and then stored overnight at −20 °C. Cells were subjected to 2 freeze-thaw cycles and the cell lysates were then centrifuged at $5,000 \times g$ for 5 min at 4 °C. Supernatants were collected and the presence of proteins were detected against MX1 (SEL763Hu), PAI1 (SEA532Hu), KNG1 (SEB267Hu), ANXA3 (SEE786Hu), FGG (SEC477Hu), MVK (SEH603Hu), ALDH6A1 (SEE836Hu) and ADH6 (SEE865Hu) from Cloud-clone, Texas, USA, while IFNGR1 (CSB-EL011051Hu) and LEAP2 (CSB-EL012853Hu) from Cusabio

Biotech, Wuhan, China. Standard curves were constructed to determine the concentration of proteins present in the samples. All analyses were done in triplicate.

## Verification of the ELISA data

To verify the ELISA results, the presence of two proteins, IFNGR1 and SERPINE1, was further assessed using Western blotting. HepG2 cells were seeded and treated as earlier described for ELISA analysis. Proteins were then extracted from cells in modified radio immunoprecipitation assay (RIPA) buffer. Briefly, the cells were washed with ice-cold PBS, trypsinized and lyzed with sufficient amount of ice-cold RIPA buffer (150 mM sodium chloride, 1% Triton X-100, 0.5% sodium deoxycholate, 0.1% sodium dodecylsulphate and 50 mM Tris–HCl at pH 8.0) containing 1% protease and phosphatase inhibitors.

Total cell lysate proteins were quantified using bicinchoninic acid (BCA) protein assay kit (Thermo Scientific, Waltham, Massachusetts, USA). Forty µg of the extracted proteins were separated on a 12% sodium dodecyl sulfate (SDS) polyacrylamide gel and transferred onto polyvinylidene difluoride (PVDF) membranes with 0.45 µm pore size (Bio-Rad, CA, USA) at a constant voltage of 100 V, 350 mA for 1 h at 4 °C in a mini electrophoretic blotting system (CBS Scientific, San Diego, California, USA). The blot was then blocked overnight and developed against anti-interferon gamma receptor 1, IFNGR1 (ab61179, Abcam, UK), anti-serpin peptidase inhibitor, clade E, member 1, SERPINE1 (ab66705, Abcam, UK) using the WesternDot 625 Goat Anti-Rabbit Western Blot kit (Invitrogen, Waltham, Massachusetts, USA). Anti-$\beta$-actin and Biotin-XX-Goat were used as the reference and secondary antibody, respectively. The membranes were first incubated with biotin-XX-Goat anti-rabbit for 1 h and then washed thrice with 1X washing buffer. Following this, the membranes were incubated with Qdot 625 streptavidin nanocrystal for 1 h and after the final washing step with deionized water, the fluorescence signal was visualized at 300 nm using a gel doc system (Gel-doc 1000/2000 system; Bio-Rad, Hercules, California, USA). The image was digitally captured, densitometrically scanned, quantified and analyzed using ImageJ Software (National Institutes of Health, Bethesda, Maryland, USA). At least three independent Western blots were quantified for each experiment.

## Statistical analyses

All analyses were done in triplicate. Results were expressed as means $\pm$ standard deviation. The data were statistically analyzed using the SPSS statistical software for Windows, Version 21.0. (IBM Corporation, Armonk, New York, USA). An independent t-test was used for comparisons of means between groups. One-way analysis of variance (ANOVA) and Tukey's Honestly Significant Difference test were used to compare means among groups. The level of significance was set at $p < 0.05$.

# RESULTS

## Cell viability assay

The viability of HepG2 cells in response to treatment with different concentrations of the methanol leaf extract of *T. indica* was determined from the log plot of cell viability assay (Fig. S1). From the log plot, the concentration of the methanol leaf extract that reduced

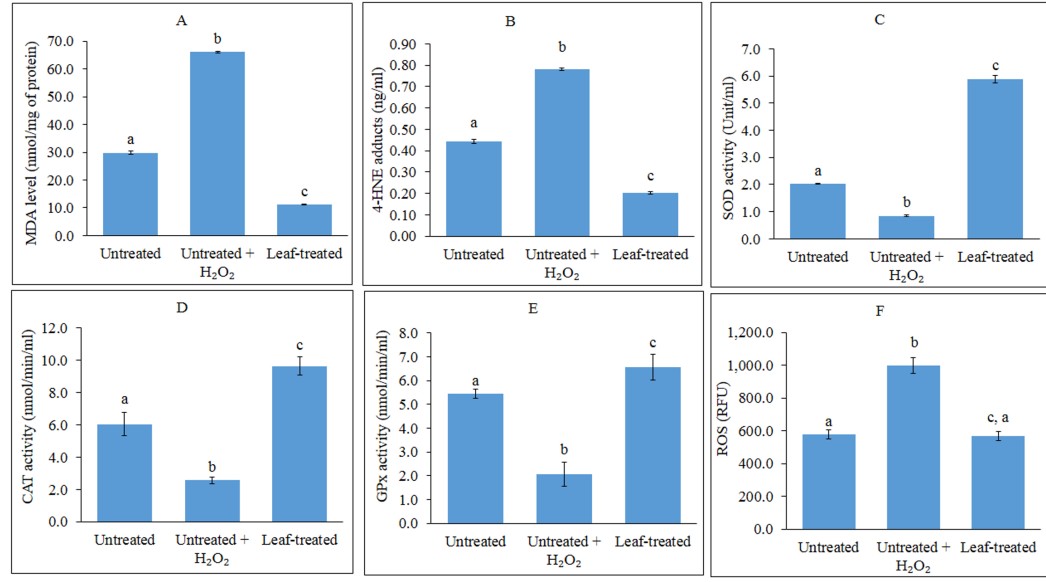

**Figure 1** **Analyses of lipid peroxidation, antioxidant enzymes and ROS.** (A–F): Analyses of lipid peroxidation, antioxidant enzymes and reactive oxygen species (ROS) in untreated, untreated + $H_2O_2$-induced and leaf-pre-treated + $H_2O_2$-induced HepG2 cells. (A–B) The effects of the leaf extract on lipid peroxidation, measured as MDA levels (A) and 4-HNE protein adduct level (B) in HepG2 cells. Results for MDA levels were expressed as nmol MDA equivalents/mg of protein while 4-HNE levels were expressed as nmol 4-HNE adduct/mg of protein. MDA-malondialdehyde; 4-HNE-4-hydroxynonenal (C–E) Superoxide dismutase (C), catalase (D) and glutathione peroxidase (E) activities were determined using commercial assay kits (Cayman Chemicals). (F) ROS production in HepG2 cells was measured using fluorescence multi-detection microplate reader and the results were expressed as Relative Fluorescence Unit (RFU). Values with different lower case letters are significantly different ($p < 0.05$).

cell viability by 50% ($IC_{50}$) was found to be 93.33 µg/ml. The highest concentration tested, 600.00 µg/ml, reduced cell viability to 2.3%. The HepG2 cells showed more than 80% viability ($IC_{20}$) in response to 24.55 µg/ml of the extract. The $IC_{20}$ concentration was subsequently used to treat the cells for the antioxidant and microarray gene expression analyses.

## Evaluation of lipid peroxidation, antioxidant enzyme activities and ROS production in HepG2 cells treated with the methanol leaf extract of *T. indica*

### Lipid peroxidation

The effects of the methanol leaf extract of *T. indica* in preventing lipid peroxidation in HepG2 cells were investigated. As shown in Fig. 1A, induction of oxidative stress by $H_2O_2$ in the untreated HepG2 cells caused the MDA levels to increase up to 3-fold higher than the untreated control cells. A significant reduction in MDA levels was observed in pre-treated cells subjected to induction of oxidative stress. ELISA analysis showed that $H_2O_2$-induced HepG2 cells displayed approximately 2-fold increase in levels of 4-HNE compared to untreated and uninduced cells (Fig. 1B). Pre-treatment of the cells with the leaf extract led to reduction of 4-HNE levels lower than the untreated, $H_2O_2$-induced cells, and even lower than the untreated, uninduced cells.

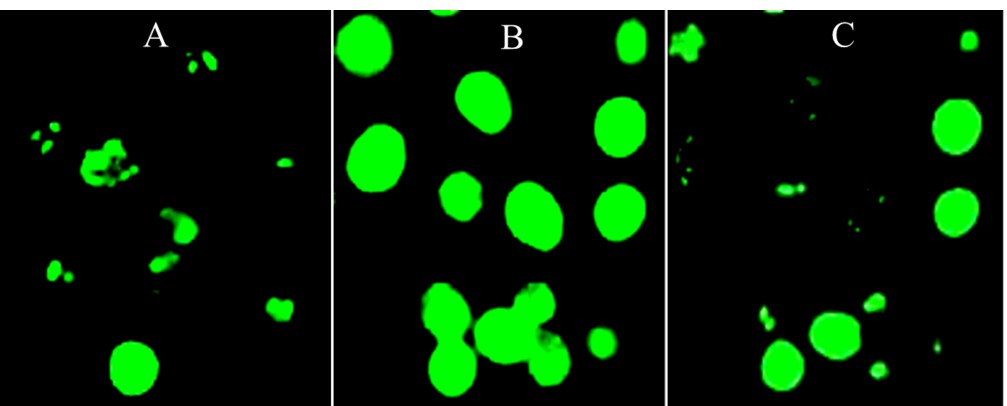

**Figure 2 Image-based ROS measurement captured by fluorescence microscopy.** (A–C) Image-based ROS measurement captured by fluorescence microscopy in HepG2 cells; (A) Untreated and unchallenged HepG2, (B) Untreated and $H_2O_2$-challenged HepG2 and (C) Leaf-pre-treated and $H_2O_2$-challenged HepG2.

## Antioxidant enzyme activities

The ability of the methanol leaf extract of T. *indica* to influence antioxidant enzyme activities were investigated by treating HepG2 cells with the $IC_{20}$ concentration of the plant extract. As shown in Figs. 1C–1E, the induction of HepG2 cells with $H_2O_2$ decreased the activities of SOD, CAT and GPx. However, all the antioxidant enzyme activities were significantly enhanced in the cells following pre-treatment with the leaf extract as compared to the untreated and untreated, $H_2O_2$-induced cells.

## Assessment of ROS production in HepG2 cells

The effect of the methanol leaf extract of *T. indica* on ROS production was evaluated in HepG2 cells using a cell permeable probe, DCF-DA. The fluorescent intensity is proportional to the amount of peroxides, which are produced by the cells. The higher the ROS amount, the higher the fluorescent intensity. In the present study, DCF-DA staining was applied to examine whether the leaf extract was able to inhibit ROS production in $H_2O_2$-induced HepG2 cells. The ROS activity was expressed as Relative Fluorescence Unit (RFU) (Fig. 1F).

## Image-based ROS measurement captured by fluorescence microscopy

ROS production increased significantly after HepG2 cells were challenged with 1 mM $H_2O_2$ as compared with the untreated and unchallenged cells (Fig. 1F). However, pre-treatment of the cells with $IC_{20}$ concentration of the leaf extract significantly decreased $H_2O_2$-mediated ROS formation, whereby intensity of the fluorescence was reduced to a level similar to that of the unchallenged cells. This can also be observed when ROS production was captured by fluorescence microscopy (Figs. 2A–2C).

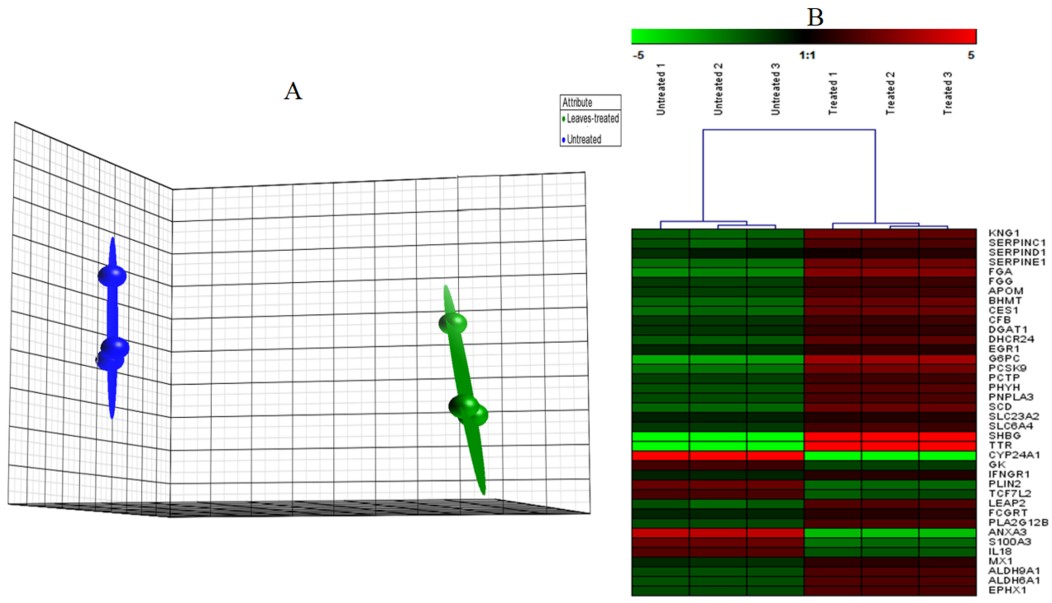

**Figure 3 PCA plot.** (A) A Principal Component Analysis (PCA) plot generated using Partek software of HepG2 cells grown in the presence or absence of the methanol leaf extract of *T. indica*. (Data are clustered based on three biological replicates ($n = 3$) of control and treated samples. Arrays for the untreated group are in blue and those for the treated group are in green. Each ball represents a sample. (B) Hierarchical clustering of highly significantly expressed genes that are associated to different pathways generated by Genesis software. The gene expression was regulated in response to the treatment with leaf extract on HepG2 cells. The regulation pattern was differentiated with 2 colours; green for down regulation and red for up regulation. The clustering was generated using Genesis software.

## Gene expression analyses in HepG2 cells treated with the methanol leaf extract of *T. indica*

### cDNA microarray data analyses

Microarray data were filtered using Partek Genomics Suite software according to *P* value less than 0.05 and with fold change difference of equal to or greater than 1.5. Principle Component Analysis (PCA) plot was then generated using the same software to indicate the reproducibility of the microarray data (Fig. 3A). Biological replicates of control ($n = 3$) and leaf-treated HepG2 cells ($n = 3$) are shown in blue and green, respectively. From the plot, it is clear that the control samples were grouped separately from the treated samples. Hierarchical clustering in Fig. 3B shows the expression pattern of selected significantly regulated genes in each replicate of the untreated and *T. indica* leaf-treated cells. Green and red boxes indicate down- and up-regulated genes, respectively. A total of 207 genes were significantly regulated in leaf-treated HepG2 cells. The complete list of significantly regulated genes is attached in Table S2. Based on GO analyses, the list of highly significant genes categorized under their biological functions with details of the GenBank accession number, name of the gene and its respective protein, and fold change difference between treated and untreated cells are presented in Table 1. Amongst the significant regulated genes, the expression of *CYP24A1* and *SHBG* showed the highest fold change difference of −4.19 and 4.11, respectively. Other significantly regulated genes include *KNG1*,

**Table 1 A list of highly significant up-regulated and down-regulated genes in HepG2 cells treated with the methanol leaf extract of *T. indica* generated using GO analyses from Partek Genomics Suite software.** The data are presented with details of the GenBank accession number, biological functions, name of the gene and its respective protein and fold change difference between treated and untreated cells. Negative values indicate down-regulation of the genes.

| GenBank ID | Protein (Gene name) | Fold change (leaf-treated vs. untreated) |
|---|---|---|
| *Hematological system development and function* | | |
| NM_000508 | Fibrinogen alpha chain (*FGA*) | 3.03 |
| NM_000602 | Serpin peptidase inhibitor, clade E, member 1 (*SERPINE1*) | 2.75 |
| NM_021870 | Fibrinogen gamma chain (*FGG*) | 2.39 |
| NM_000488 | Serpin peptidase inhibitor, clade C, member 1 (*SERPPINC1*) | 2.14 |
| NM_000893 | Kininogen 1 (*KNG1*) | 1.84 |
| NM_000185 | Serpin peptidase inhibitor, clade D, member 1 (*SERPIND1*) | 1.58 |
| *Lipid metabolic and transport process* | | |
| NM_001713 | Betaine-homocysteine S-methyltransferase (*BHMT*) | 2.65 |
| NM_174936 | Proprotein convertase subtilisin/kexin type 9 (*PCSK9*) | 2.39 |
| NM_025225 | Patatin-like phospholipase domain containing 3 (*PNPLA3*) | 2.31 |
| NM_005063 | Stearoyl-CoA desaturase (delta-9-desaturase) (*SCD*) | 2.17 |
| NM_001025195 | Carboxylesterase 1 (*CES1*) | 2.13 |
| NM_014762 | 24-dehydrocholesterol reductase (*DHCR24*) | 1.91 |
| NM_001964 | Early growth response 1 (*EGR1*) | 1.83 |
| NM_019101 | Apolipoprotein M (*APOM*) | 1.80 |
| NM_001045 | Solute carrier family 6 (*SLC6A4*) | 1.76 |
| NM_005116 | Solute carrier family 23 (*SLC23A2*) | 1.71 |
| NM_012079 | Diacylglycerol O-acyltransferase homolog 1 (*DGAT1*) | 1.68 |
| NM_006214 | Phytanoyl-CoA 2-hydroxylase (*PHYH*) | 1.63 |
| NM_001102402 | Phosphatidylcholine transfer protein (*PCTP*) | 1.54 |
| NM_001710 | Complement factor B (*CFB*) | 1.52 |
| NM_001122 | Perilipin 2 (*PLIN2*) | −2.33 |
| *Carbohydrate metabolic process* | | |
| NM_000151 | Glucose-6-phosphatase, catalytic subunit (*G6PC*) | 2.50 |
| NM_001128127 | Glycerol kinase (*GK*) | −2.27 |
| NM_001146274 | Transcription factor 7-like 2 (T-cell specific, HMG-box) (*TCF7L2*) | −1.56 |
| *Regulation of hormone* | | |
| NM_001040 | Sex hormone-binding globulin (*SHBG*) | 4.11 |
| NM_000371 | Transthyretin (*TTR*) | 3.40 |
| *Inflammatory response* | | |
| NM_032562 | Phospholipase A2, group XIIB (*PLA2G12B*) | 1.50 |
| NM_002960 | S100 calcium binding protein A3 (S100A3) | −3.05 |
| NM_001562 | Interleukin 18 (*IL18*) | −2.39 |

Table 1 (*continued*)

| GenBank ID | Protein (Gene name) | Fold change (leaf-treated vs. untreated) |
|---|---|---|
| *Defense response to virus/bacterium* | | |
| NM_052971 | Liver expressed antimicrobial peptide 2 (*LEAP2*) | 1.96 |
| NM_002462 | Myxovirus (influenza virus) resistance 1 (*MX1*) | 1.58 |
| NM_000416 | Interferon gamma receptor 1 (*IFNGR1*) | 1.51 |
| NM_005139 | Annexin A3 (ANXA3) | −3.32 |
| *Xenobiotic metabolic process* | | |
| NM_000120 | Epoxide hydrolase 1 (*EPHX1*) | 2.12 |
| NM_005589 | Aldehyde dehydrogenase 6 family, member A1 (*ALDH6A1*) | 1.86 |
| NM_000850 | Glutathione S-transferase mu 4 (*GSTM4*) | 1.83 |
| NM_000696 | Aldehyde dehydrogenase 9 family, member A1 (*ALDH9A1*) | 1.67 |
| NM_001102470 | Alcohol dehydrogenase 6 (*ADH6*) | 1.65 |
| NM_000782 | Cytochrome P450, family 24, subfamily A, polypeptide 1 (*CYP24A1*) | −4.19 |
| *Immune response* | | |
| NM_004107 | Fc fragment of IgG, receptor, transporter (*FCGRT*) | 1.62 |

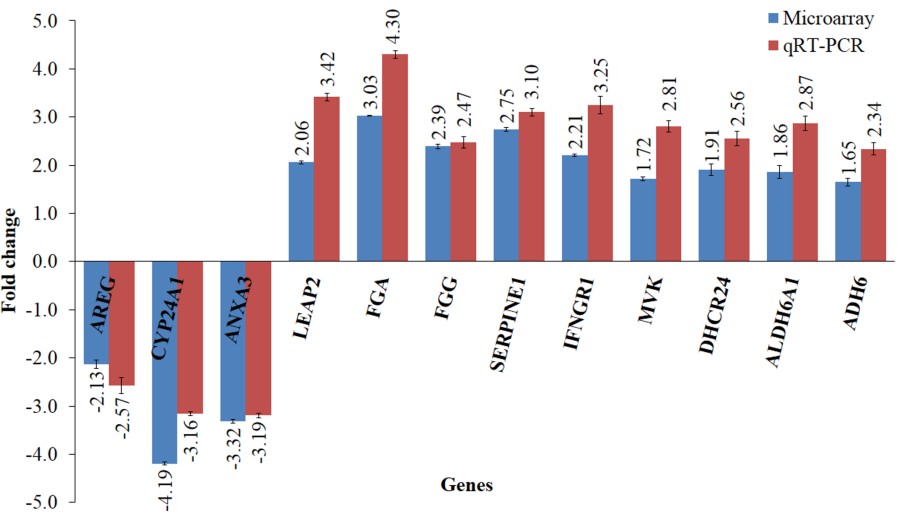

**Figure 4 Validation of microarray data using qRT-PCR.** The bar chart shows the gene expression patterns (presented as fold change) of selected significantly regulated genes calculated using qRT–PCR and microarray analysis. The down-regulated genes selected were *CYP24A1, ANXA3* and *AREG*, while the up-regulated genes were *LEAP2, FGA, FGG, SERPINE1, IFNGR1, MVK, DHCR24, ALDH6A1* and *ADH6*. All qRT-PCR data were normalized to that of *GAPDH*, a housekeeping gene.

*SERPINC1, SERPIND1, SERPINE1, FGG, FGA, DHCR24, LEAP2, IFNGR1, ANXA3, MX1, ADH6* and *ALDH6A1* (Table 1). Similar pattern of expression was generated when selected genes namely *AREG, CYP24A1, ANXA3, FGG, FGA, LEAP2, SERPINE1, MVK, DHCR24, IFNGR1, ADH6* and *ALDH6A1* were quantitated relative to that of *GADPH*, using qRT-PCR (Fig. 4).

**Table 2   IPA analysis.** A summary of Ingenuity Pathway Analysis (IPA) showed the top networks and top predicted canonical pathways affected by the altered expression of genes in response to the treatment of the leaf extract in HepG2 cells.

| ID | Top networks | Score[a] |
|----|--------------|----------|
| 1 | Lipid metabolism, small molecule biochemistry, hematological disease | 36 |
| 2 | Ophthalmic Disease, Connective Tissue Disorders, Inflammatory Disease | 35 |
| 3 | Digestive System Development and Function, Organ Morphology, Developmental Disorder | 32 |
| 4 | Carbohydrate Metabolism, Small Molecule Biochemistry, Free Radical Scavenging | 24 |
| 5 | Hereditary Disorder, Neurological Disease, Organismal Injury and Abnormalities | 24 |

| Top canonical pathway | P-value | Ratio | List of gene |
|-----------------------|---------|-------|--------------|
| Coagulation system | $2.80 \times 10^{-6}$ | 6/38 (0.158) | *KNG1, SERPINC1, SERPIND1, SERPINE1, FGG, FGA* |
| Superpathway of cholesterol biosynthesis | $2.17 \times 10^{-4}$ | 4/87 (0.046) | *DHCR24, LSS, MVK, TM7SF2* |
| Intrinsic prothrombin activation pathway | $2.92 \times 10^{-4}$ | 4/37 (0.108) | *KNG1, SERPINC1, FGA, FGG* |
| Immune protection/ antimicrobial response | $2.28 \times 10^{-3}$ | 4/109 (0.037) | *LEAP2, IFNGR1, ANXA3, MX1* |
| Xenobiotic metabolism signaling | $2.41 \times 10^{-3}$ | 6/304 (0.020) | *ALDH6A1, ALDH9A1, EPHX1, CYP24A1, ADH6, GSTM4* |

**Notes.**
[a] A score of 2 or higher indicates at least 99% confidence of not being generated by random chance and higher scores indicate a greater confidence.

## Pathway interactions and biological process analyses of the significantly expressed genes

IPA analyses identified "Lipid Metabolism, Small Molecule Biochemistry and Hematological Disease" as the top putative network linking 21 of the significantly regulated genes with other interactomes, with a score of 36 (Table 2). A score of 2 or higher indicates confidence of at least 99% of not being generated by random chance and higher scores indicate greater confidence. "Ophthalmic Disease, Connective Tissue Disorders, Inflammatory Disease" was ranked second with a score of 35 while "Digestive System Development and Function, Organ Morphology, Developmental Disorder" was placed third with a score of 32. Two other networks; "Carbohydrate Metabolism, Small Molecule Biochemistry, Free Radical Scavenging" and "Hereditary Disorder, Neurological Disease, Organismal Injury and Abnormalities" shared a similar score of 24. Figure 5A shows a graphical representation of the predicted molecular relationships between the genes listed under the top networks, which linked them to the top five predicted canonical pathways that are listed in Table 2. IPA identified "Lipid Metabolism, Small Molecule Biochemistry and Hematological Disease" as the top network that linked the genes of altered expression namely *KNG1, SERPINC1, SERPIND1, SERPINE1, FGG* and *FGA*, to the two top canonical pathways, "Coagulation System" ($P < 2.80 \times 10^{-6}$) and "Intrinsic Prothrombin Activation Pathway" ($P < 2.92 \times 10^{-4}$). This network also linked *MVK, DHCR24, LSS* and *TM7SF2* to the second top canonical pathway, "Superpathway of Cholesterol Biosynthesis"

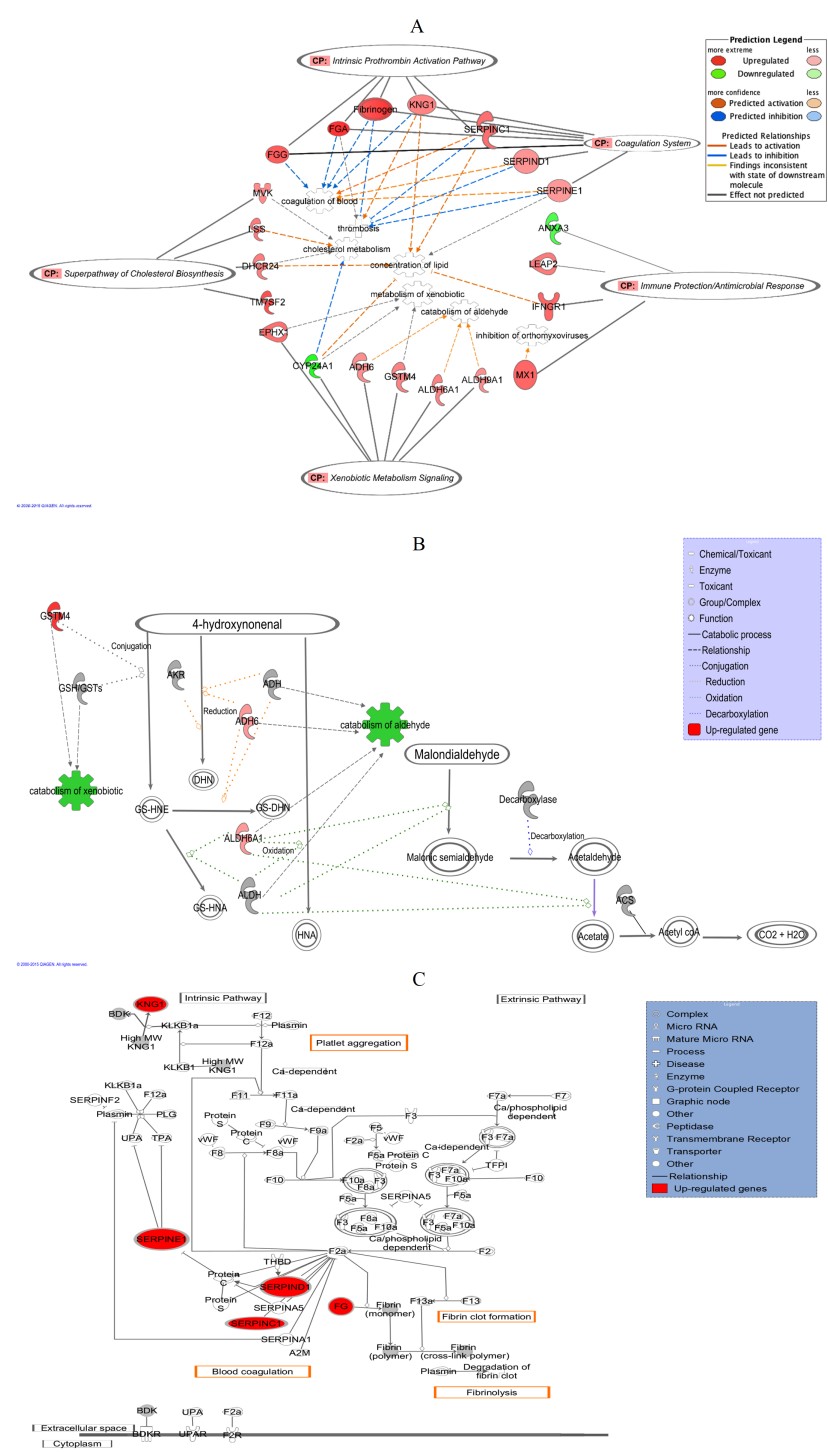

**Figure 5 IPA graphical representation.** (A) IPA graphical representation of the molecular relationships between the significantly regulated genes to the predicted canonical pathways in response to the leaf treatment in HepG2 cells. The network is displayed graphically as nodes (genes) and edges (the biological relationships between the nodes). Nodes in red indicate 

**Figure 5 (...continued)**

up-regulated genes while those in green represent down-regulated genes. Various shapes of the nodes represent functional class of the proteins. Edges are displayed with various labels that describe the nature of the relationship between the nodes. Name of genes with their corresponding abbreviations are as follows: *ADH6*, Alcohol dehydrogenase 6; *ALDH6A1*, Aldehyde dehydrogenase 6 family, member A1; *ALDH9A1*, Aldehyde dehydrogenase 9 family, member A1; *ANXA3*, Annexin A3; *CYP24A1*, Cytochrome P450, family 24, subfamily A, polypeptide 1; *DHCR24*, 24-dehydrocholesterol reductase; *EPHX1*, Epoxide hydrolase 1; *FGA*, Fibrinogen alpha chain; *FGG*, Fibrinogen gamma chain; *GSTM4*, Glutathione S-transferase mu 4; *IFNGR1*, Interferon gamma receptor 1; *KNG1*, Kininogen 1; *LEAP2*, Liver-expressed antimicrobial peptide; *LSS*, Lanosterol synthase; *MVK*, Mevalonate kinase; *MX1*, Myxovirus resistance 1; *SERPINC1*, Serpin peptidase inhibitor, clade C (antithrombin), member 1; *SERPIND1*, Serpin peptidase inhibitor, clade D (heparin cofactor), member 1, *SERPINE1*, Serpin peptidase inhibitor, clade E (Nexin, Plasminogen activator inhibitor, type 1), member 1; *TM7SF2*, Transmembrane 7 superfamily member 2. (B) IPA graphical representation showing the effect of significantly regulated genes, *ALDH6A1*, *ALDH6* and *GSTM4* in the detoxification process of 4-HNE and MDA. This figure demonstrates that the three genes are involved in the xenobiotic metabolism signaling pathway as on one of the top canonical pathway generated by IPA. The process is displayed graphically as nodes (genes) and edges (the biological relationships between the nodes). Nodes in red indicate up-regulated genes. Various shapes of the nodes represent functional class of the proteins. Edges are displayed with various labels that describe the nature of the relationship between the nodes. (C) IPA graphical representation of the molecular relationships between *KNG1*, *SERPINE1*, *SERPINC1*, *SERPIND1* and Fibrinogen that are involved in "Coagulation System", the top predicted canonical pathway in HepG2 cells affected by the methanol leaf extract of *T. indica*. The network is displayed graphically as nodes (genes) and edges (the biological relationships between the nodes). Nodes in red indicate up-regulated genes. Various shapes of the nodes represent functional class of the proteins. Edges are displayed with various labels that describe the nature of the relationship between the nodes. Name of genes/proteins with their corresponding abbreviations are as follows: *A2M*, Alpha-2-macroglobulin; *BDK*, Bradykinin; *BDKR*, Bradykinin receptor; F2, Coagulation factor II (thrombin); F2a, Coagulation factor IIa (thrombin); F2R, Coagulation factor II receptor; F3, Coagulation factor III (thromboplastin, tissue factor); F5a, Coagulation factor V (proaccelerin, labile factor); F7, Coagulation factor VII (serum prothrombin conversion accelerator); F7a, Coagulation factor VIIa (serum prothrombin conversion accelerator); F8, Coagulation factor VIII (procoagulant component); F8a, Coagulation factor VIIIa (procoagulant component); F9, Coagulation factor IX; F9a, Coagulation factor IXa; F10, Coagulation factor X; F10a, Coagulation factor Xa; F11, Coagulation factor XI; F11a, Coagulation factor XIa; F12, Coagulation factor XII (Hageman factor); F12a, Coagulation factor XIIa (Hageman factor); F13, Coagulation factor XIII; F13a, Coagulation factor XIIIa; *KLKB1a*, Kallikrein B plasma 1a; *PLG*, Plasminogen; *SERPINA1*, Serpin peptidase inhibitor, clade A, member 1; *SERPINA5*, Serpin peptidase inhibitor, clade A, member 5; *SERPINF2*, Serpin peptidase inhibitor, clade F, member 2; *TFPI*, Tissue factor pathway inhibitor; *THBD*, Thrombomodulin; *TPA*, Tissue plasminogen activator; *UPA*, Urokinase plasminogen activator; *UPAR*, Urokinase plasminogen activator receptor; vWF, von Willebrand factor.

($P < 2.17 \times 10^{-4}$). On the other hand, both top networks, "Ophthalmic Disease, Connective Tissue Disorders, Inflammatory Disease" and "Digestive System Development and Function, Organ Morphology, Developmental Disorder" linked *LEAP2*, *IFNGR1*, *ANXA3* and *MX1* to the fourth top canonical pathway "Immune Protection/Antimicrobial Response" ($P < 2.28 \times 10^{-3}$). The next top network, "Carbohydrate Metabolism, Small Molecule Biochemistry, Free Radical Scavenging" connected *ALDH6A1*, *ALDH9A1*, *EPHX1*, *CYP24A1*, *ADH6* and *GSTM4* to the fifth top canonical pathway, "Xenobiotic Metabolism Signaling" ($P < 2.41 \times 10^{-3}$).

Figure 5B shows the IPA graphical representation of *ALDH6A1*, *ADH6* and *GSTM4* in the xenobiotic metabolism signaling particularly in the major detoxification process of 4-HNE and MDA, the toxic byproducts from oxidative stress-induced lipid peroxidation. On

the other hand, Fig. 5C shows the IPA graphical representation of the interaction between *SERPINC1*, *SERPIND1*, *SERPINE1*, *KNG1*, *FGG* and *FGA* with other interactomes in the "Coagulation System," the top predicted canonical pathway that were affected by methanol leaf extract of *T. indica*. Four of the genes, *KNG1*, *SERPINC1*, *FGG* and *FGA*, were also linked to the third top canonical pathway, "Intrinsic Prothrombin Activation Pathway."

## Detection of altered protein abundance in HepG2 cells treated with the methanol leaf extract of *T. indica* using ELISA and Western blotting

Figure 6A shows significantly higher levels of ALDH6A1, ADH6, IFNGR1, LEAP2, SERPINE1, MX1, KNG1, MVK and FGG in the cells treated with the antioxidant-rich methanol leaf extract of *T. indica* compared to the untreated cells. On the other hand, ANXA3 level was significantly lower in the treated cells compared to those untreated. Western blot analysis showed that after normalization with $\beta$-actin, SERPINE1 and IFNGR expression were increased by 2.02 and 2.47-fold, respectively, in the treated cells (Fig. 6B).

## DISCUSSION

Different parts of *T. indica* are recognized for their various medicinal properties. We have reported earlier that the antioxidant-rich *T. indica* fruit pulp extract was able to significantly regulate the expression of a sizable number of genes (*Razali, Aziz & Junit, 2010*; *Lim et al., 2012*) and significantly alter protein abundance (*Chong et al., 2012*; *Chong et al., 2013*) that are associated with antioxidant activities and lipid metabolisms in HepG2 cells. In addition, our group has recently reported that the leaves of *T. indica* possessed high phenolic contents and had potent antioxidant activities in scavenging free radicals in non-cellular based assays (*Razali et al., 2012*). Catechin, epicatechin and quercetin were detected in the antioxidant-rich leaf extract of *T. indica* (*Razali et al., 2012*). Catechin and epicatechin isolated from cocoa showed potent antioxidant activities (*Gu et al., 2006*). Catechin and epicatechin detected in tea prevented oxidative stress in rats' liver by directly altering the subcellular ROS production, glutathione metabolism and cytochrome P450 2E1 activity (*Higdon & Frei, 2003*; *Sang et al., 2003*; *Singh, Shankar & Srivastava, 2011*). However, nutrigenomic analyses to further evaluate the molecular mechanism associated with the medicinal and health benefits of the leaves of *T. indica* in HepG2 cells have not been widely conducted and reported.

HepG2 cells were pre-treated with the antioxidant-rich methanol leaf extract of *T. indica*, followed by the induction of oxidative damage with $H_2O_2$. This was done to investigate the ability of the antioxidant-rich leaf extract of *T. indica* to enhance antioxidant protection in the cells, hence preventing them from oxidative damage. The cells were not exposed to $H_2O_2$ prior to treatment with the plant extracts as our aim was to measure the ability of *T. indica* to prevent oxidative damage in the cells, rather than repairing damage after it has occurred.

In this study, the antioxidant-rich methanol leaf extract of *T. indica* was shown to be able to reduce the production of ROS and reduce the levels of malondialdehyde (MDA) levels and 4-HNE-protein adducts in HepG2 cells.

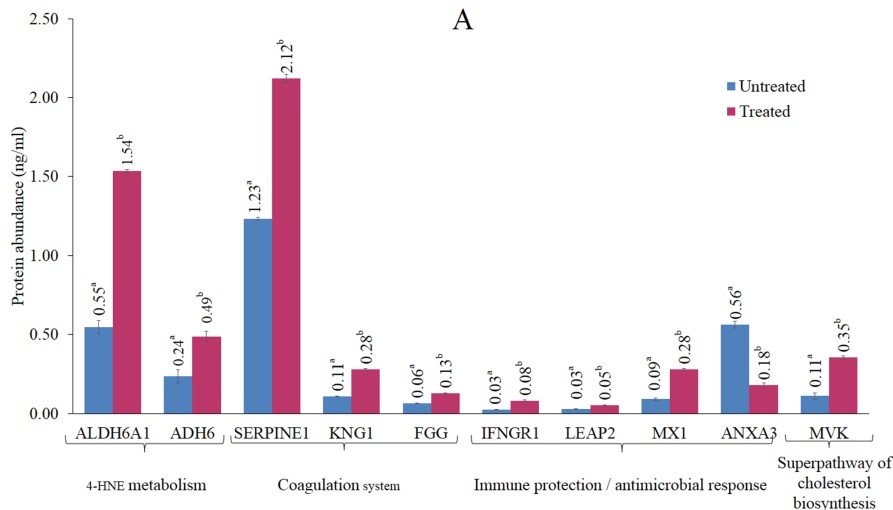

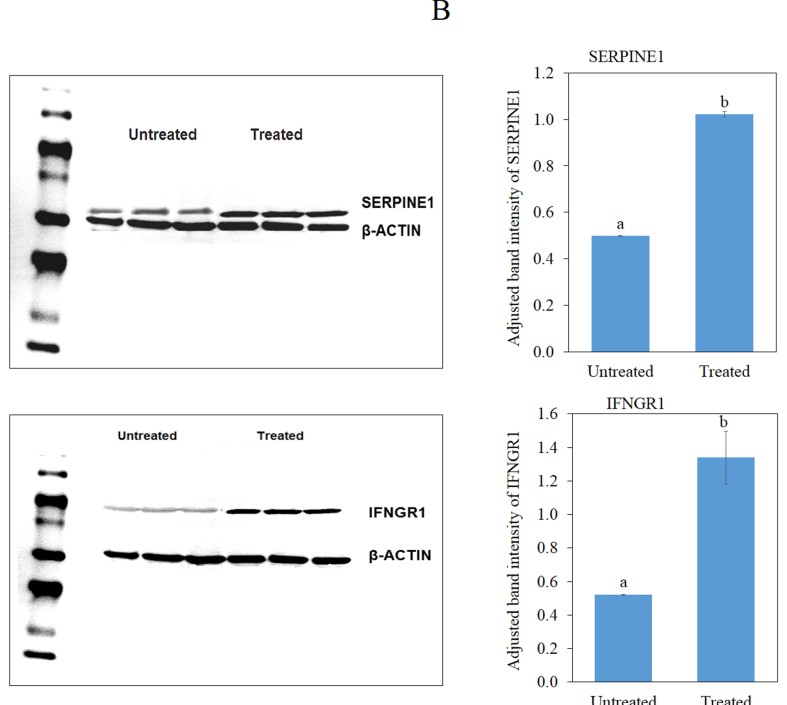

**Figure 6 ELISA analyses of selected proteins.** (A) Enzyme-Linked Immunosorbent Assay (ELISA) analyses of the human IFNGR1, LEAP2, SERPINE1, ANXA3, KNG1, MX1, FGG, MVK, ALDH6A1 and ADH6 antibodies level in the untreated and leaf-treated HepG2 cells. ELISA analyses were done according to manufacturer's protocols (Cloud-clone, Houston, Texas, USA; Cusabio Biotech, Wuhan, China). Bars not sharing the same superscript letter indicate significant difference at $p < 0.05$ (B) Up regulation of IFNGR1 (a) and SERPINE1 (b) after treatment with $IC_{20}$ concentration of the leaf extract for 24 h. Protein levels were measured with specific antibodies by western blot analysis; $\beta$-actin was the loading control. Untreated cells were used as control. The experiments were repeated in triplicates and the representative blot was shown. Bars not sharing the same superscript letter indicate significant difference at $p < 0.05$.

Lipid peroxidation is known to be an important factor in the pathology of many diseases associated with oxidative stress. Interaction between ROS and unsaturated lipids especially polyunsaturated fatty acids (PUFAs) forms hydroperoxides which can then be further metabolized to produce a variety of aldehydes, including MDA (*Brattin, Glende & Recknagel, 1985*) and 4-HNE (*Esterbauer & Cheeseman, 1990*; *Spickett, 2013*). MDA and 4-HNE were found to be elevated in various diseases (*Negre-Salvayre et al., 2008*; *Reed, 2011*) and are widely used as biomarkers for lipid peroxidation (*Zhang et al., 2008*). DCF-DA is a fluorogenic dye that measures hydroxyl, peroxyl and other ROS activity within the cell. After diffusion into the cell, DCF-DA is deacetylated by cellular esterases to a non-fluorescent compound, dichloro-dihydro fluorescein (DCFH). In the presence of peroxidase, intracellular $H_2O_2$ or OH· changes DCFH to the highly fluorescent compound DCF (*Oh & Lim, 2006*). $H_2O_2$ is produced from microsomes and peroxisomes and also by several physiological processes such as inflammatory respiratory burst and oxidative phosphorylation (*Valko et al., 2006*; *Chen et al., 2011*). $H_2O_2$ has often been used as a model to investigate the mechanism of cell injury by oxidative stress (*Rigoulet, Yoboue & Devin, 2011*). From this study, it can be speculated that the antioxidant-rich leaf extract might also mediate protection of HepG2 cells against toxicity towards $H_2O_2$ by the inhibition of hepatic lipid peroxidation, supporting previous report by *Ghoneim & Eldahshan (2012)*. MDA (and HNE) is/are enzymatically catabolized to $CO_2$ and $H_2O$ by mitochondrial aldehyde dehydrogenase (ALDH), decarboxylase and acetyl CoA synthase or by cytoplasmic phosphoglucose isomerase to methyoglyoxal (MG) and subsequently to D-lactate. HNE can form protein or DNA adducts resulting in further biomolecule damages. Based on the microarray analyses, genes encoding glutathione transferase (GST), ALDH and alcohol dehydrogenase (ADH) were significantly up-regulated. All these proteins were involved in the major detoxification pathways to convert 4-HNE and MDA to less reactive chemical species and control their steady-state intracellular concentrations (*Spickett, 2013*). 4-HNE metabolism may lead to the formation of corresponding alcohol 1,4-dihydroxy-2-nonene (DHN), acid 4-hydroxy-2-nonenoic acid (HNA), and HNE–glutathione conjugate products (*Dalleau et al., 2013*). The main catabolic reactions of 4-HNE are the formation of 4-HNE adducts with glutathione (GSH), which occurs spontaneously or can be catalyzed by GSTs. 4-HNE conjugation with GSH produces glutathionyl-HNE (GS-HNE) followed by NADH-dependent ADH catalyzed its reduction to glutathionyl-DHN (GS-DHN) and/or ALDH catalyzed oxidation to glutathionyl-HNA (GS-HNA). Secondly, 4-HNE can be reduced to DHN by aldo-keto reductases (AKRs) or ADH. Thirdly, 4-HNE can be oxidized to HNA by ALDH (*Zhong & Yin, 2015*). On the other hand, MDA metabolism involves its oxidation by mitochondrial ALDH followed by decarboxylation to produce acetaldehyde, which is then oxidized by ALDH to acetate and further to $CO_2$ and $H_2O$ (*Esterbauer, Schaur & Zollner, 1991*). As shown in Fig. 5B, the up-regulation of *ALDH6A1*, *ADH6* and *GSTM4* in the leaf-treated HepG2 cells affected the major detoxification process of 4-HNE and MDA. When analyzed using IPA, all these genes were associated with the top canonical pathway, "Xenobiotic Metabolism Signaling." The findings could explain the potent antioxidant activities of the leaf extract in the previous assays mainly in inhibiting lipid peroxidation. From the ELISA results, the level of 4-HNE adducts were found to be

reduced after the leaf treatment as compared to the untreated HepG2 cells. This was in agreement with the previous analyses which demonstrated the reduced level of MDA in the treated HepG2 cells. In addition, the encoded proteins levels, ALDH6A1 and ADH6, were also increased in response to the leaf extract treatment suggesting that protection against lipid peroxidation in the liver was through an increased degradation of MDA and HNE.

SOD, GPx and CAT are three of the primary antioxidant enzymes in mammalian cells that are important in oxygen metabolizing cells. These antioxidant enzymes are regarded as the first line of defense against ROS. SODs catalyze the dismutation of superoxide anion radicals into $H_2O_2$ and water, while CAT and GPx convert $H_2O_2$ into water, reducing the amounts of $H_2O_2$, hence protecting the cells against oxidative damage (*McCord, Keele & Fridovich, 1971*). The plant extract was shown to increase the activities of SOD, CAT and GPx, further reducing levels of the radicals, superoxide anion and $H_2O_2$, hence lowering peroxidative damage to lipids. However, the increase in enzyme activities was not corroborated with an increase in expression of the encoding genes suggesting that the leaf extract did not directly influence gene transcription but rather, by increasing the activity of the enzymes.

Results in this study imply that the leaf extract which is rich in phenolics including catechin and epicatechin (*Razali et al., 2012*) may have the ability to directly scavenge ROS and/or free radicals that are produced endogenously in the HepG2 cells. This observation may also explain the potent activity of the leaf extract in inhibiting lipid peroxidation and increasing the antioxidant enzyme activities in the previous results. In the event when MDA and HNE accumulate in cells, the leaf extract was capable to regulate the expression of the genes and consequently the proteins that are involved in the degradation of the two lipid peroxidation products, MDA and HNE, to ultimately produce $CO_2$ and $H_2O$.

In addition to the genes encoding GST, ALDH and ADH, a total of 207 genes were significantly regulated when HepG2 cells were exposed to an $IC_{20}$ concentration of the antioxidant-rich leaf extract of *T. indica*. Comparable pattern was observed when the expression of selected genes including *AREG, CYP24A1, ANXA3, FGA, FGG, SEPRINE1, MVK, DHCR24, IFNGR1 LEAP2, ALDH6A1* and *ADH6* were quantitated using qRT-PCR, validating the results obtained from the microarray analyses (*Morey, Ryan & Van Dolah, 2006*). When the significantly regulated genes were subjected to IPA analyses, "Lipid Metabolism, Small Molecule Biochemistry, Hematological Disease" was identified as the top biological network associated with the significantly regulated genes with a score of 36. In addition, the "Coagulation System" was identified as the top canonical pathway linking 6 genes namely *SERPINC1, SERPIND1, SERPINE1, KNG1, FGA* and *FGG* that were mainly associated with anticoagulation and coagulation cascades in the coagulation system (Fig. 5C). *SERPINC1, SERPIND1, SERPINE1* and *KNG1* encode anti-thrombin III (ATIII), heparin cofactor II (HCII), plasminogen activator inhibitor-1 (PAI-1) and high molecular weight kininogen (HMWK) protein, respectively, while both *FGA* and *FGG* encode for fibrinogen. The top networks suggests that the methanol leaf extract of *T. indica* affect the top canonical pathways by altering the expression of those genes, converging mainly on coagulation of blood, cholesterol and xenobiotic metabolisms and antimicrobial response.

ATIII belongs to the serpin peptidase inhibitor (SERPIN) family, the largest and most diverse family of protease inhibitors (*Rawlings, Tolle & Barrett, 2004*). ATIII cooperates with its co-factor, heparin, to effectively regulate coagulation proteases, such as thrombin and coagulation factors, Xa, IXa, XIa, and XIIa, (*Siddiqi, Tepler & Fantini, 1997*) thus, prevents the activation of clotting proteinases in blood except at the site of a vascular injury (*Olson et al., 2010*). Deficiencies in ATIII correlated with an increased risk of thrombosis (*Siddiqi, Tepler & Fantini, 1997*) and CVD (*Brouwer et al., 2009*). Certain polyphenolic antioxidants namely quercetin (standard quercetin or extracted from plants) showed anticoagulant activities by enhancing ATIII levels (*Mozzicafreddo et al., 2006*; *Hsieh et al., 2007*). The leaf extract of *T. indica* have been reported to contain quercetin (*Razali et al., 2012*), suggesting its role in enhancing the expression of the *SERPINCI* gene, which may subsequently affect the ATIII synthesis and secretion.

ROS on the other hand can mediate intravascular thrombus formation by interfering with mechanisms that normally inhibit activation of the coagulation pathway, thus promoting the pro-thrombic cascade (*Gorog & Kovacs, 1995*). Lipid peroxides for instance, can increase the amount of thrombin produced and can slow down the rate of thrombin decay (*Salvemini & Cuzzocrea, 2002*). Hence, the ability of the leaf extract of *T. indica* to inhibit lipid peroxidation and prevent ROS generation, together with its high antioxidant activities may be beneficial in providing protection against the development of thrombus.

*SERPIND1* encodes for another anticoagulant, heparin cofactor II (HCII). HCII potently inhibits thrombin action by forming a bimolecular complex with dermatan sulfate or heparin proteoglycans under the endothelial layer in mammalians (*Tollefsen, Pestka & Monafo, 1983*). HCII shares a 30% amino acid sequence homology to that of ATIII (*Baglin et al., 2002*). Unlike ATIII, HCII exclusively inhibits thrombin and does not inhibit other proteases that are involved in coagulation or fibrinolysis (*He et al., 2002*). HCII interacts with dermatan sulfate in the vessel wall after disruption of the endothelium and this interaction regulates thrombus formation thus maintaining blood flow after injury to the arterial endothelium (*He et al., 2002*). Defects in *SERPIND1*, leading to HCII deficiency are the cause of thrombophilia, a haemostatic disorder characterized by a tendency to recurrent thrombosis, causing a hypercoagulation state (*Tollefsen, Pestka & Monafo, 1983*). Quercetin, isolated from *Flaveria bidentis*, was reported to enhance HCII levels (*Guglielmone et al., 2002*). Quercetin, which was also found in high amount in the leaf extract of *T. indica* (*Razali et al., 2012*), may be responsible for the reported anticoagulation property of the extract (*Mozzicafreddo et al., 2006*; *Guglielmone et al., 2002*) by regulating the expression of both *SERPINCI* and *SERPIND1*.

In addition to *SERPINCI* and *SERPIND1*, the expression of *SERPINE1, KNG1, FGA* and *FGG* genes of which the encoded proteins are involved in coagulation cascade, were also significantly up-regulated in HepG2 cells in response to the methanol leaf extract of *T. indica*. PAI-1 also shares a 30% amino acid sequence homology to that of ATIII. HMWK, upon binding with prekallikrein, initiates the intrinsic pathway of the coagulation cascade and through a series of events, modulates fibrin clot structure.

Fibrinogen plays several key roles in the maintenance of hemostasis (*Lang et al., 2009*) and clot formation (*He et al., 2003*; *Bolliger, Gorlinger & Tanaka, 2010*). Fibrinogen

molecules act during both cellular and fluid phases of coagulation. In the cellular phase, it facilitates the aggregation of platelets via binding of glycoprotein IIb/IIIa receptors on platelet surfaces (*Mosesson, 2005*). In the fluid phase, it is cleaved by thrombin to produce fibrin monomers, which polymerize to form the basis of the clot thus, provide the structural network required for effective clot formation (*Tanaka, Key & Levy, 2009*). Fibrinogen also functioned as an acute phase reactant, *in vivo* to help modulate the inflammatory cellular reactions (*Levy et al., 2012*).

The fibrin clot formation from fibrinogen is dissolved by plasmin. Plasmin is proteolytically released from plasminogen by tissue plasminogen activator (tPA) (*Kjalke et al., 2001*). The activity of tPA is selectively inhibited by PAI-1 (*Stringer et al., 1994*). PAI-1 binds to polymerized fibrin within the thrombus, followed by inhibition of tPA-mediated fibrinolysis (*Loskutoff & Samad, 1998*). The absence of PAI-1 in humans leads to life-long bleeding problems presumably resulting from the development of hyper fibrinolysis (*Loskutoff & Quigley, 2000*). Therefore, PAI-1 is the primary physiological inhibitor of plasminogen activation *in vivo*, and elevations in plasma PAI-1 appear to promote normal fibrin clearance mechanisms and promote thrombosis (*Loskutoff & Samad, 1998*). PAI-1 is encoded by *SERPINE1* which expression was up-regulated in response to the methanol leaf extract of *T. indica* treatment in this study. Previous studies reported that a green tea polyphenol, epigallocatechin-3-gallate, suppressed the expression of tPA, of which, was selectively inhibited by PAI-1 (*Maeda-Yamamoto et al., 2003*; *Ramos, 2008*). It is speculated that epicatechin detected in the leaf extract might be involved in regulating the *SERPINE1* expression, thus may possibly inhibit tPA expression. Hence, the presence of both epicatechin and quercetin might possibly contribute towards the anticoagulant properties of the methanol leaf extract of *T. indica*. Four genes, *KNG1*, *SERPINC1*, *FGG* and *FGA*, were also associated to the "Intrinsic Prothrombin Activation Pathway" further suggesting that the leaf extract was capable in regulating the coagulation cascade probably involving the intrinsic prothrombin activation pathway.

Excessive formation of blood clots may cause acute thrombus formation and obstruction, thus lead to a series of cardiovascular disease (CVD) complications such as stroke. On the other hand, deficiency in blood clotting might cause prolonged bleeding and led to the development of hypovolemic shock, impaired consciousness, acute renal failure, hypoxia, and progressive respiratory failure. This shows the importance of hemostasis in the coagulation cascade in order to achieve a desired balanced in clot production. Patients with CVD are often given anticoagulants as one of the medications, to prevent or delay blood coagulation and the formation of blood clots. However, many of these drugs, namely heparin, dermatan sulphate and warfarin, carry the risk of prolonged bleeding. Antithrombin inhibits factor Xa, thrombin and other serine proteases. These reactions are relatively slow but are markedly accelerated by anticoagulant drugs binding to antithrombin through a high-affinity pentasaccharide sequence, which produces a conformational change in antithrombin that increases antifactor Xa and antithrombin activity (*Gresele & Agnelli, 2002*). This results in a hypocoagulable state in patients under these anticoagulant drug treatments, which is associated with approximately 20–30%

of patients' mortality (*Franchini, 2005*). Therefore, the ideal clinical anticoagulant would reliably and predictably inhibit thrombin without substantially increasing the risk of bleeding. A recent study revealed that tamarind leaves did not cause significant haemolysis in human red blood cells (*Escalona-Arranz et al., 2014*). In the present study, the antioxidant-rich leaf extract of *T. indica* appeared to target various genes including those in intrinsic prothrombin activation pathway that could contribute towards the regulation of the coagulation cascade and thus provides an alternative approach to the classical anticoagulants. However, further in-depth *in vitro* and *in vivo* analyses are required to verify the findings as well as to ensure its safety and efficacy.

The methanol leaf extract of *T. indica* also appeared to affect the "Superpathway of Cholesterol Biosynthesis" by regulating the expression of *DHCR24*, *LSS*, *MVK* and *TM7SF2* genes. Validation using qRT-PCR also showed a consistent alteration of *DHCR24* and *MVK*. Oxidative stress leads to the production of ROS which can attack lipid membrane constituents such as unsaturated phospholipids, glycolipids, and cholesterol, resulting in cellular dysfunction and cell death (*Girotti, 1998*). Studies have reported that plasma membrane compartments rich in cholesterol may participate in signal transduction pathways activated upon oxidative stress, and thus enhance pro survival pathways, while cholesterol depletion appears to increase apoptosis in the oxidative stress-induced cells (*Yang, Oo & Rizzo, 2006*). During oxidative stress, the expression of genes involved in lipid and cholesterol homeostasis, was shown to be activated by caspase cleavage (*Yokoyama et al., 1993*). It has been reported that increased cholesterol content in cells concomitant to enhanced antioxidant capacity, made them less susceptible to oxidative stress (*Kolanjiappan, Ramachandran & Manoharan, 2003*). These observations have led to the hypothesis that cholesterol is required as protection to the cells during oxidative damage to maintain plasma membrane integrity (*Di Stasi et al., 2005*).

*MVK* encodes for mevalonate kinase which phosphorylates mevalonate to mevalonate 5-phosphate in the cholesterol biosynthetic pathway. Apart from cholesterol as the end-product, the pathway is also responsible in synthesising several non-sterol isoprenes such as ubiquinone-10 or also known as coenzyme Q10 (CoQ10) (*Goldstein & Brown, 1990*). Studies have reported the ability of CoQ10 to act as hydrophilic antioxidants and to protect biological membrane against lipid peroxidation (*Ernster & Forsmark-Andree, 1993*). In fact, ubiquinol-10 was reported to be more efficient in preventing peroxidative damage to LDL than the well-known antioxidants lycopene, beta-carotene, or alpha-tocopherol (*Stocker, Bowry & Frei, 1991*). The up-regulation of *MVK* in this study could have led to the synthesis of lipid-soluble CoQ10 that might have contributed to the lipid peroxidation inhibitory effect of the antioxidant-rich leaf extract of *T. indica*.

The antioxidant-rich leaf extract of *T. indica* also significantly regulated the expression of *DHCR24* that encodes for 24-dehydrocholesterol reductase (DHCR24), the enzyme that catalyzes the reduction of delta-24 double bond of sterol intermediates during cholesterol biosynthesis (*Waterham et al., 2001*). DHCR24 has been shown to modulate membrane cholesterol levels and lipid raft formation (*Crameri et al., 2006*). Lipid rafts are cholesterol-rich microenvironments on the cell surface. They are required for cell func-

tions, including directed mobility and capping of membrane proteins, receptor-mediated signalling, entry and exit of pathogens and membrane trafficking (*Simons & Toomre, 2000*). DHCR24 has been reported to act as antioxidants via two mechanisms; the first through a cholesterol-dependent manner (*Kuehnle et al., 2008*), possibly involving pro survival factors such as Akt and the second through direct scavenging of $H_2O_2$ (*Lu et al., 2008*). DHCR24 was originally known as Seladin-1 gene (Selective Alzheimer's Disease Indicator-1) as its expression was initially discovered to be down-regulated in regions of the brain vulnerable to Alzheimer's disease (AD) (*Iivonen et al., 2002*). Amyloid-$\beta$ (A$\beta$) peptide accumulation in the central nervous system underlies the pathological process in AD. Pharmacological enhancement of DHCR24 activity was reported to be protective against A$\beta$ toxicity and oxidative stress-induced apoptosis in AD brain (*Greeve et al., 2000*).

Lanosterol synthase (*LSS*) catalyzes the cyclization of (S)-2,3-oxidosqualene to form lanosterol during sterol biosynthesis (*Ausubel et al., 1999*). Oxidative stress has been reported to increase the level of lanosterol in mitochondria and several other intracellular compartments of macrophages, suggesting that this sterol metabolite may be part of a global cellular response to stress (*Andreyev et al., 2010*).

More recently, defects in regulation of sterol metabolism has also been implicated in Parkinson's disease (PD). *Lim et al. (2012)* reported that lanosterol showed neuroprotective effect in dopaminergic neurons in various models of PD. From this study, it is speculated that the leaf extract of *T. indica* may provide antioxidative protection against ROS-induced oxidative damage that can lead to AD and PD for instance, by regulating the expression of *DHCR24*, *MVK* and *LSS* genes of the cholesterol biosynthesis superpathway. Studies reported the neuroprotective effects of citrus (*Hwang, Shih & Yen, 2012*) and blackcurrant (*Vepsäläinen et al., 2013*) flavonoids against A$\beta$ neurotoxicity by up-regulating the expression of genes involved in cholesterol metabolic pathway including *DHCR24* (*Hwang, Shih & Yen, 2012*). Amongst the flavonoids are quercetin and myricetin. Hence, quercetin together with the other flavonoids detected in the leaf extract might possibly work by a similar mechanism.

The leaf extract of *T. indica* has been reported to have antimicrobial activities (*Meléndez & Capriles, 2006*) against some common gram negative and gram positive bacteria namely *E. coli* (*Meléndez & Capriles, 2006*) and *Burkholderia pseudomallei* (*Muthu, Nandakumar & Rao, 2005*). Polyphenols detected in the methanol leaf extract of *T. indica* were known to exhibit antioxidative potential and have many beneficial effects on human health. Previously, in addition to quercetin and epicatechin, other polyphenols including isorhamnetin, were also detected in the leaf extract of *T. indica* (*Razali et al., 2012*). Isorhamnetin and quercetin isolated from speckled alder (*Alnus incana* ssp. *rugosa*) and the leaf extract of *Calotropis procera* (*Nenaah, 2013*) were reported to possess antimicrobial activity against Gram-positive bacteria including *Staphylococcus aureus* and Gram-negative bacteria namely *E. coli* (*Rashed et al., 2014*). Therefore, isorhamnetin and quercetin may have contributed towards the reported antimicrobial properties of the antioxidant-rich leaf extract of *T. indica*, through direct action on the expression of antimicrobial response-related genes, *IFNGR1*, *LEAP2*, *ANXA3* and *MX1*.

Interferon-$\gamma$ (IFN-$\gamma$) is a key mediator of the host immune response, and the IFN-$\gamma$ receptor 1 subunit that was encoded by *IFNGR1* is essential for IFN-$\gamma$ binding and signalling (*Bach, Aguet & Schreiber, 1997*). IFN-$\gamma$ induces the expression of various cytokines and chemokines during the course of viral infection, including HBV and HCV (*Barber, 2001*), and thus induces the virus clearance by cellular innate responses to eliminate viruses (*Korachi et al., 2013*).

*LEAP2* codes for liver-expressed antimicrobial peptide-2 (LEAP-2), which significantly shares structural characteristics with antimicrobial peptides such as defensins and LEAP-1/hepcidin. LEAP-2 is the second blood-derived peptide that is expressed predominantly in the liver and exhibits antimicrobial activity. As the first blood-derived antimicrobial peptide from the liver, LEAP-1/hepcidin (*Krause et al., 2003*; *Park et al., 2001*) has been reported to be involved in iron homeostasis (*Nicolas et al., 2001*; *Pigeon et al., 2001*). These important physiological functions might be possible for LEAP-2 as well. A previous study has shown that LEAP-2 displayed dose-dependent antimicrobial activity against numerous Gram-positive bacteria and yeasts namely *S. cerevisiae* and *B. subtilis* (*Krause et al., 2003*). In fact, a study by *Howard et al. (2010)* elucidated a role for this peptide in protecting HepG2 cells from injury and/or stress.

Myxovirus resistance protein (MX) encoded by *MX1* plays a major role in IFN-induced host defense. MX, once induced by IFN-$\alpha$ and $\beta$, blocks viral replication cycle at an early stage thus provided an early antiviral innate immunity (*Haller, Frese & Kochs, 1998*). Human MX demonstrates a wide antiviral spectrum of activities against orthomyxo viruses including influenza viruses, rhabdoviruses, Bunyaviridae and paramyxoviride (*Frese et al., 1996*; *Tumpey et al., 2007*). Numerous experiments have also indicated that recovery from virus infection in humans requires a functional MX defense system (*Staeheli et al., 1986*; *Moritoh et al., 2009*).

Annexins are a family of proteins that binds to phospholipids and membranes in a calcium-dependent manner and may link calcium signalling to membrane functions (*Larsson et al., 1997*). *ANXA3* also exhibited important roles in tumor development, metastasis and drug resistance (*Wu et al., 2013*) and showed up-regulated expression in many active cancers (*Hayes & Moss, 2004*; *Mussunoor & Murray, 2008*). Furthermore, it has been shown that *ANXA3* interacted with molecules of the S100 family of calcium binding molecules (*Gerke & Moss, 2002*). Members of this family are suggested to play an important role in psoriasis pathogenesis and showed up-regulated expression in psoriatic lesional skin, thus suggesting its pathogenic role in this disease (*Koczan et al., 2005*).

Changes in the expression of a gene may not necessarily lead to alteration in the abundance of the corresponding protein. However, in this study, it was evident that the alteration in gene expression by HepG2 when exposed to the leaf extract was corroborated by changes in abundance of the corresponding proteins. The antioxidant-rich leaf extract of *T. indica* appeared to be able to alter the expression of proteins that are involved in the Coagulation System and the Intrinsic Prothrombin Activation Pathway (KNG1, SERPINE1, FGG), Superpathway of Cholesterol Biosynthesis (MVK), Immune protection/antimicrobial response (IFNGR1, LEAP2, ANXA3 and MX1) and Xenobiotic

Metabolism Signaling (ALDH6A1, ADH6). Western blot analysis showed consistent pattern of expression of the SERPINE1 and IFNGR1 was detected hence validating the ELISA and microarray data.

## CONCLUSION

The antioxidant-rich methanol leaf extract of *T. indica* extract showed protective effects in HepG2 cells by inhibiting lipid peroxidation, enhancing the antioxidant enzyme activities and suppressing the ROS production. The present study implies the potential of this plant as an alternative source of a natural antioxidant agent and the promising therapeutic potential of the leaves of *T. indica*. Inhibition of lipid peroxidation involved alteration in the expression of genes and the encoded proteins which are responsible for the degradation of MDA and HNE. The antioxidant-rich methanol leaf extract of *T. indica* extract also directly targeted the expression of genes and the encoded proteins that are involved in the coagulation system and antimicrobial response hence providing molecular evidence associated to the medicinal properties of the leaf extract.

## ACKNOWLEDGEMENTS

We would like to thank Siah Eng Tian and Teng Loong Hung from Research Instruments Malaysia for their technical expertise.

### Funding

This research project was funded by research grants (PV116-2012A and H-20001-00-E000009-B29000) from University of Malaya, Kuala Lumpur, Malaysia and FP015-2013B from the Ministry of Education, Malaysia. The funders had no role in study design, data collection and analysis, decision to publish, or preparation of the manuscript.

### Grant Disclosures

The following grant information was disclosed by the authors:
University of Malaya: PV116-2012A, H-20001-00-E000009-B29000.
Ministry of Education, Malaysia: FP015-2013B.

### Competing Interests

The authors declare there are no competing interests.

### Author Contributions

- Nurhanani Razali performed the experiments, analyzed the data, wrote the paper, prepared figures and/or tables, reviewed drafts of the paper.
- Azlina Abdul Aziz and Sarni Mat Junit conceived and designed the experiments, analyzed the data, contributed reagents/materials/analysis tools, wrote the paper, reviewed drafts of the paper.
- Chor Yin Lim performed the experiments, analyzed the data.

## Microarray Data Deposition

The following information was supplied regarding the deposition of microarray data:

GEO-NCBI: http://www.ncbi.nlm.nih.gov/geo/query/acc.cgi?acc=GSE71606.

## Supplemental Information

Supplemental information for this article can be found online at http://dx.doi.org/10.7717/peerj.1292#supplemental-information.

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
