# Peer review of "Investigation into the effects of antioxidant-rich extract of Tamarindus indica leaf on antioxidant enzyme activities, oxidative stress and gene expression profiles in HepG2 cells"

_PeerJ, doi:10.7717/peerj.1292_

## Round 0.1 · original submission · Major Revisions

You need to address all the critical points raised by both reviewers.

PeerJ does not offer copyediting, please ensure that the English language in this submission meets journal's standards: uses clear and unambiguous text, is grammatically correct, and conforms to professional standards of courtesy and expression.

Reviewer 1 ·

Basic reporting

No comments

Experimental design

No comments.

Validity of the findings

No comments.

Additional comments

Comments to the Author:

The manuscript entitled “Investigation into the effects of antioxidant-rich extract of Tamarindus indica leaf on antioxidant enzyme activities, oxidative stress and gene expression profiles in HepG2 cells” by Razali et al. is a brief-report that describes a very interesting study of relevance for new drugs development community.
The group shows that the natural antioxidant-rich extract derived from leaves of Tamarindus indica, inhibited lipid peroxidation and ROS production, enhanced antioxidant enzyme activities and significantly regulated the expression of genes and proteins involved with consequential impact on the coagulation system, cholesterol biosynthesis, xenobiotic metabolism signaling and antimicrobial response. The manuscript is very well written and I have read it with great enthusiasm. In my opinion it can be accepted with minor corrections, since it is an interesting paper, have good and extensive methodology and discussion, but the authors should just clarify some details during the manuscript. I have some specific comments with some concerns and minor suggestions to further improve the manuscript.

Specific comments



Page6
Have you for instance performed any experiments, making the exposure of H2O2 before the treatment with the Tamarindus indica? If not, maybe in the discussion section you should discuss that. Speculate/Comment what would you expect.

Page7
Line153 – I suggest to add that you have added secondary antibodies and HRP conjugates.

Page9
Line 191 – How many cells were used in these experiments? Did you use the same 96 wells plates?

Page 16
Line 336: Lipid Peroxidation – Figure 1
Refering to figure 1, I miss the description of the statistics between cells that received pre-treatment with the extract and peroxide (second bar) with the cells that were exposed to peroxide without extract (third bar). Did you forget the statistic symbol or is it true that there is no difference between them? Because it looks apparently that in some graphs the second and third groups have differences. And the symbol “a” appears in some figures where it doesn’t look to have any difference (for instance, figure 1f: leaf-treated x untreated). Please, take a close attention to this figure.
Also, do you have a group that was pre-treated with the leaves but have no H2O2 induction?

Page 18
Line 375: Please clarify here which was the experimental condition.
Are you comparing which groups?
When I read your sentence:

Page 21
Line 442: Please discuss why your group decided to study the effects of the leaves in HepG2 cells.

Page24
Line 504: substitute were for was

Discussion
Personally, I thought it was a long discussion, but very complete. Even though I think you should add at least some comments on the possible components of the extract that might be responsible for the effects seen.

Reviewer 2 ·

Basic reporting

Title: Investigation into the effects of antioxidant-rich extract of Tamarindus indica leaf on antioxidant enzyme activities, oxidative stress and gene expression profiles in HepG2 cells

Major points:
There are many reports of extracts of T. indica recently, which should be included in the Introduction part of the manuscripts, such as those of Nakchat O, 2015; Escalona-Arranz JC, 2014; and Agnihotri A, 2013. In Materials and methods, the time of harvest of the plants should be indicated in the “Preparation part” of the methanol leaf extract of T. indica. In each method, there should be reference(s), some do not contain such as measurement of HNE-protein adduct and quantification of antioxidant enzyme activities. In Results: Figure 2, the data were qualitative and subjective as shown by micrographs. DCF fluorescence intensity should be replaced with quantitative data (such as being measured by the fluorescence intensity of the tested group compared to control and reporting as folds of fluorescence intensity using fluorescence plate reader or flow cytometry). In Figure 4, is it significant between the data of gene expressions measured by microarray and those of qRT-PCR? In Figure 5a-5c, the letters and/or the color lines and/or symbols in the boxes in the right upper corner, are not clear enough to be readable, this should be magnified. In Figure 6a, there was no “a” (indicates significant difference) in the top of bar graphs in Figure6a as mentioned by its legend. In Figure 6b, is it significant in the data (T. indica leaf extract-treated group) compared to the control? Why did the authors choose these two proteins to be verified further by Western blot in Figure 6b? In Results, the subtitles were inappropriate. The Discussion part was not focus and concise, it should be pointed out to the target(s) or objective(s). There are several paragraphs and descriptive data which are too long and un-relevant to each other in this Discussion part. The authors did not give their rationales or comments and the application of this study and for the linkages between pathways for clinical use. Line 505, the authors did not give any evidence to support that the gene transcription of antioxidant enzymes did not alter. There is a published paper by the same author group in 2010 Razali N, about the same herb study of gene expression profile in human HepG2 but using the pulp of the plant whereas in this study the leaf is used. The authors should give some comparisons or discussion in these findings.

Minor points:
The full name should be given at the first time it occurred then followed by the abbreviated form, such as CVD and DCFH-DA. The reference format is not consistent with each other, such as Line 792, the journal name is in full name whereas others in abbreviated forms. There are many grammar and typo errors that should be carefully corrected. The plant species should be in italic, especially in the References, such as Line 843.

Experimental design

No comments

Validity of the findings

Figure 2 for the measurement of ROS is qualitative and subjective, which should be quantitative by using fluorescence plate reader or flow cytometry.

Additional comments

No comments.

Annotated reviews are not available for download in order to protect the identity of reviewers who chose to remain anonymous.

---

## Round 0.2 · accepted · Accept

Thank you for addressing all comments of the reviewers and for careful editing and revising your manuscript.